# Thermodynamic Analysis of a Flat Plate Solar Collector with Different Hybrid Nanofluids as Working Medium—A Thermal Modelling Approach

**DOI:** 10.3390/nano13081320

**Published:** 2023-04-09

**Authors:** R. M. Mostafizur, M. G. Rasul, M. N. Nabi, R. Haque, M. I. Jahirul

**Affiliations:** 1Fuel and Energy Research Group, School of Engineering and Technology, Central Queensland University, Rockhampton, QLD 4702, Australia; 2Fuel and Energy Research Group, School of Engineering and Technology, Central Queensland University, Melbourne, VIC 3000, Australia; 3School of Science, Technology and Engineering, University of the Sunshine Coast, Sippy Downs, QLD 4556, Australia

**Keywords:** flat plate solar collector, hybrid nanofluid, entropy generation, exergy efficiency, heat transfer coefficient, pumping power

## Abstract

In this study, the performance of hybrid nanofluids in a flat plate solar collector was analysed based on various parameters such as entropy generation, exergy efficiency, heat transfer enhancement, pumping power, and pressure drop. Five different base fluids were used, including water, ethylene glycol, methanol, radiator coolant, and engine oil, to make five types of hybrids nanofluids containing suspended CuO and MWCNT nanoparticles. The nanofluids were evaluated at nanoparticle volume fractions ranging from 1% to 3% and flow rates of 1 to 3.5 L/min. The analytical results revealed that the CuO-MWCNT/water nanofluid performed the best in reducing entropy generation at both volume fractions and volume flow rate when compared to the other nanofluids studied. Although CuO-MWCNT/methanol showed better heat transfer coefficients than CuO-MWCNT/water, it generated more entropy and had lower exergy efficiency. The CuO-MWCNT/water nanofluid not only had higher exergy efficiency and thermal performance but also showed promising results in reducing entropy generation.

## 1. Introduction

The demand for investment in sustainable energy has grown in recent years due to climate change caused by the excessive use of fossil fuels [1]. Global warming has been increasing since the beginning of industrialisation. This is due to the increased production of glasshouse gases. Both the deterioration of the environment and the benefits of human progress come with high costs that humanity must bear. Now, it is very necessary to make use of alternative or renewable sources of energy to slow the progression of climate change and slow the consumption of fossil fuels [2]. Renewable energy is depleting; solar energy is a solid solution due to its eco-friendly behaviour. As a result of this, it is being considered as a potential alternative energy source to mitigate the adverse consequences of energy sources to lessen the impact of fossil fuels [3].

Collecting and concentrating the sun’s radiation into usable heat energy requires special equipment known as solar collectors and concentrators. These collectors are utilised to generate power as well as heat the working fluids. The stored heat energy from radiation is absorbed and transported by the working fluid. Utilizing a thermal device like a flat-plate solar collector (FPSC) to heat using absorption solar energy is cost-effective and environmentally beneficial [4]. Air, water, ethylene glycol, and oil heat transfer fluids absorb solar energy from the FPSC. The FPSC takes in energy from the sun and sends it to a conventional heat transfer fluid including air, water, ethylene glycol, and oil. The FPSC is a highly versatile solar collector due to its simple design, low manufacturing and maintenance costs, and its ability to be widely used in low-temperature water heating and process heat applications. Even researchers [5,6] are trying to integrate solar thermal energy storage attached to the bottom of the collectors. In addition, the suspension of new and innovative materials into conventional fluids is being evaluated to increase the thermal efficiency of FPSCs [7]. The traditional fluids are mixed with suspended nano-sized (1–100 nm) metal or metal oxide particles, which are known as nanofluids [8].

One of the advantages of using nanofluids as a working fluid is an enhancement in thermal conductivity, which enhances the working fluid’s ability to heat transfer capacity. Additionally, nanofluids with the appropriate particle volume and size can improve solar energy collecting efficiency [9]. As a result, the FPSC’s thermal efficiency is improved. The large surface area to volume ratio of nanoparticles in the nanofluids is their main advantage [10]. When it comes to the thermal performance of solar collector systems, increasing heat transfer, reducing pressure drops, optimising the ratio of heat transfer to the pressure drop, and reducing costs are crucial criteria for judging designs. In addition to these requirements, Bejan [11] stated that entropy generation from a system should be considered as another crucial factor, and that it is best to aim to reduce the overall rate of entropy generation. According to Bejan [12], lowering total entropy creation lowers a system’s irreversibility, or energy loss. The use of hybrid nanomaterials, a new concept of nanofluids, and the usage of individual nanomaterials at a higher level, helps to maximise some properties and minimise other impacts necessary for applications [13]. In the base fluid, more than one type of nanoparticle is suspended together to make these hybrid nanofluids (hnf). The hybrid material combines the thermal and rheological characteristics of both materials at the same time, providing improved properties in a homogeneous phase over mono nanofluids [14]. When used in solar collectors, hybrid nanofluids aim to improve the working fluid’s thermal and rheological characteristics, allowing for more efficient absorption of solar radiation at reduced costs [15].

Numerous academics are looking at flat plate solar collectors as a potential source of renewable energy and mitigating their potential environmental problems [16,17]. The use of nanofluids as an absorbing medium for solar energy collecting and increasing the efficiency of solar plate collectors has received significant attention in recent years. Otanicar et al. [18] conducted an experiment to determine the extinction index of four liquids in solar collectors: water, ethylene glycol, propylene glycol, and therminol VP-1. Water has shown to be the most effective solar energy absorber among all four liquids, absorbing around 13% of the total energy, which is still regarded as quite low. The most popular nanoparticles are Al_2_O_3_ and CuO which are used to prepare nanofluids for solar collectors. The thermal conductivity of the CuO nanofluids in the solar collector was greater than that of other nanoparticles, according to the analysis reported by Alim et al. [19]. Lu et al. [20] studied an open thermosyphon utilising a CuO water-based nanofluid in a solar collector to investigate the thermal performance. CuO-H_2_O nanofluid improved the evaporator’s thermal efficiency and evaporating heat transfer coefficient by around 30% when compared to water. When Al_2_O_3_ nanoparticles were utilised as the working fluid by Yousefi et al. [21] in place of water, they demonstrated an increase in efficiency of up to 28.3%. Saidur et al. [22] looked at the possibility of employing Al_2_O_3_-H_2_O nanofluid in direct absorption solar collectors. They concluded that 1 vol% of Al_2_O_3_-H_2_O significantly increased solar absorption. Tiwari et al. [23] experimentally examined the impact of utilising Al_2_O_3_-H_2_O nanofluid as an absorbing medium in a flat plate solar collector. They claimed that adding 1.5 vol% fractions of Al_2_O_3_ to a water-based nanofluid increased efficiency by 31.64% and reduced carbon dioxide emissions per kWh while using a solar collector in hybrid mode.

Recently, some research has been carried out to evaluate the usefulness of employing hybrid nanofluids in solar collectors. A summary of new experiments using hybrid nanofluids in solar collectors was provided by Suman et al. [24]. According to the study, the use of hybrid nanofluids would be crucial in future energy applications. The prospect of using these fluids to capture and store renewable energy in the future is quite encouraging. Researchers looked at the characteristics of hybrid nanoparticles made of SiO_2_/Ag and Multi-Walled Carbon Nanotubes (MWCNTs) for solar applications [25]. The outcomes showed that the working fluid’s ability to transfer energy was improved by the addition of MWCNT nanoparticles.

The heat transfer performance of the ETSC was tested by utilising mono and hybrid nanofluids of CuO and Cu on a water basis by the researchers in [26]. According to their findings, introducing CuO/Cu hybrid nanofluid increased thermal–optical performance by 61.7% and 14.9%, respectively, over water and CuO/water nanofluid. In the same study, the solar collector was submitted to economic analysis; the findings revealed that employing CuO/Cu hybrid nanofluid had shortened the Simple Payback Period (SPP) by 19.6%. Using a hybrid nanofluid made of Fe_3_O_4_ and MWCNTs, Cui et al. [27] investigated the energy effectiveness of the ETSC. According to the findings, hybrid nanofluid improved the collector’s thermal efficiency by 28.3% and 14% over water and Fe_3_O_4_/water nanofluid, respectively. Solar collectors can benefit from the use of hybrid nanofluids if the right pair of nanomaterials is chosen. On the performance of solar collectors, Farajzadeh et al. [28] looked at the numerical and experimental impacts of utilising a hybrid TiO_2_/Al_2_O_3_ nanofluid. The utilisation of TiO_2_, Al_2_O_3_, and their mixtures, according to the data, led to improvements in the efficiency of 19%, 21%, and 26%, respectively. The collector’s efficiency has also increased by 5%. CuO/MWCNTs and MgO/MWCNTs hybrid nanofluids were used in an experiment by Verma et al. [29] on a flat plate solar collector. They came to the conclusion that the performance of a flat plate solar collector with MgO hybrid nanofluid is significantly better than that of the CuO hybrid-based nanofluid.

According to the literature [30], researchers from various parts of the world have carried out experimental research and shown that nanofluids may enhance the efficiency of FPSCs. In order to assess the efficacy of solar thermal systems based on FPSCs, it is crucial to create simulation tools that use the theoretical model of FPSCs and the available techniques to estimate the parameters of nanofluids. Some recent studies focused on investigating the thermal performance based on different weather conditions and water-based hybrid nanofluids [31,32,33,34]. Therefore, it can be found from the literature review that there is no study available based on different base fluids-based hybrid nanofluids. To the author’s best knowledge, different base fluids-based hybrid nanofluids have not been included in the theoretical model analysis of FPSCs yet. In particular, not enough focus has been paid to the research on the use of water, ethylene glycol, methanol, radiator coolant and engine oil-CuO/MWCNT hybrid nanofluids in FPSCs.

The main objective of the study is to modify the theoretical FPSC model so that it can operate with water, ethylene glycol, methanol, radiator coolant and engine oil -CuO/MWCNT hybrid nanofluids. Additionally, many models that have been utilised in the literature to calculate the thermophysical characteristics of nanofluids were examined in this study. The analysis has been presented for five types of hybrids nanofluids made of CuO/MWCNT nanoparticles mixed with water, ethylene glycol, methanol, radiator coolant, and engine oil. CuO volume fractions of 1% to 3% and nanofluid flow rates of 1 to 3.5 L/min are considered for the analysis.

## 2. Theoretical Background

### 2.1. Thermal Modeling

The collector’s absorber plate absorbs energy from the sun’s radiation and then transfers the absorbed heat to a fluid, which serves as the working medium in the heat transfer process. To capture as much solar energy as possible, the highly absorbent surface of the absorber plate is designed to allow the solar radiation to pass through the transparent glass cover and strike the plate. To minimize heat loss due to wind, any stationary air between the glass cover and the heated absorber plate is minimized. Additionally, this minimizes radiative heat loss from the collector by ensuring that the wavelength released by the heated plate is in the 3–100 μ range. A nanofluid running through the pipes absorbs a significant amount of the heat energy received by the hot absorber plate. Figure 1 illustrates the construction of the Flat Plate Solar Collector (FPSC). Information about the dimensions and features of the FPSC in Table 1.

### 2.2. Entropy Analysis

The laws of thermodynamics are explained and elaborated to determine the exergy efficiency of the system in this section. The following assumptions were made for this analysis:

➢The system is a steady flow and steady-state conditions.➢No chemical reaction was presents.➢Changes of potential and kinetic energy is ignored.➢Solar irradiation intensity, *I_T_* = 1000 W/m^2^, and the absorbed power density by the solar panel, *S = I_T_(τα)* was considered.➢The sun is considered as a reservoir of energy. As stated by Takashima [36], the exergy flow rate in the global solar radiation ${\dot{E}}_{X,Sun}$ is equal to the solar flux ${\dot{Q}}_{sun}{}_{,in}=A_{C}I_{T}={\dot{E}}_{X,Sun}$➢The thermo-physical properties of the nanofluids, both entering and leaving from the collector, are constant.➢Work transfer from the system and heat transfer to the system are positive. ➢Loss coefficient is only considered for the entrance effect.

If the changes in potential and kinetic energy are ignored, the typical exergy rate stabilities can be given by [37];
(1)$$\sum \dot{E}x_{in}-\sum \dot{E}x_{out}=\sum \dot{E}x_{dest}$$
or
(2)$$\dot{E}x_{heat}-\dot{E}x_{work}-\dot{E}x_{mass,in}-\dot{E}x_{mass,out}=\dot{E}x_{dest}$$

The general exergy balance can be expressed by
(3)$$\sum \left(1-\frac{T_{a}}{T_{sur}}\right){\dot{Q}}_{s}-\dot{W}+\sum {\dot{m}}_{in}\Psi_{in}-\sum {\dot{m}}_{out}\Psi_{out}=\dot{E}x_{dest}$$
where,
(4)$$\Psi_{in}=\left(h_{in}-h_{a}\right)-T_{a}\left(s_{in}-s_{a}\right)$$
(5)$$\Psi_{out}=\left(h_{out}-h_{a}\right)-T_{a}\left(s_{out}-s_{a}\right)$$

If Equations (4) and (5) are inserted in Equation (3), it can rearrange as
(6)$$\sum \left(1-\frac{T_{a}}{T_{sur}}\right){\dot{Q}}_{s}-\dot{m}\left[\left(h_{out}-h_{in}\right)-T_{a}\left(s_{out}-s_{in}\right)\right]=\dot{E}x_{dest}$$

The solar energy rate absorbed by the collector absorber surface is represented by ${\dot{Q}}_{s}$, which can be calculated by using the expression [38] shown in Equation (7):(7)$${\dot{Q}}_{s}=I_{T}\left(\tau \alpha \right)A_{c}=S\cdot A_{c}$$

The deviations in enthalpy and entropy of the nanofluid inside the collector can be expressed by [37]
(8)$$\Delta h=h_{out}-h_{in}=C_{p,hnf}\left(T_{f,out}-T_{f,in}\right)$$
(9)$$\Delta s=s_{out}-s_{in}=C_{p,hnf}\ln\frac{T_{f,out}}{T_{f,in}}-R\ln\frac{P_{out}}{P_{in}}$$

Replacing Equations (7)–(9) in Equation (6), it can be rewritten as
(10)$$\left(1-\frac{T_{a}}{T_{sur}}\right)I_{T}\left(\tau \alpha \right)A_{c}-\dot{m}C_{p,hnf}\left(T_{f,out}-T_{f,in}\right)+\dot{m}C_{p,hnf}T_{a}\ln\frac{T_{f,out}}{T_{f,in}}-\dot{m}RT_{a}\ln\frac{P_{out}}{P_{in}}=\dot{E}x_{dest}$$

The exergy destruction (or irreversibility) rate $\dot{E}x_{dest}$ can be appraised using Equation (11):(11)$$\dot{E}x_{dest}=T_{a}{\dot{S}}_{gen}$$

Bejan [12] has provided a formula for calculating the overall rate of entropy generation (*S_gen_*) in a non-isothermal solar FPSC, which is given below.
(12)$${\dot{S}}_{gen}=\dot{m}C_{p,hnf}\ln\frac{T_{out}}{T_{in}}-\frac{{\dot{Q}}_{s}}{T_{s}}+\frac{{\dot{Q}}_{o}}{T_{a}}$$

The right-hand side of Equation (12) consists of two terms. The first term represents the non-isothermal heat transfer to the working fluid between the inlet and outlet, while ${\dot{Q}}_{s}$ and ${\dot{Q}}_{o}$ represents the rate of solar radiation heat transfer and heat loss rate to the ambient, respectively. An energy balance equation can be used to determine the total heat loss rate to the ambient can be given as,
(13)$${\dot{Q}}_{o}={\dot{Q}}_{s}-\dot{m}C_{p,hnf}\left(T_{out}-T_{in}\right)$$

The second law (or exergy) efficiency can be computed as,
(14)$$\eta_{II}=1-\frac{T_{a}{\dot{S}}_{gen}}{\left[1-\left(T_{a}/T_{s}\right)\right]{\dot{Q}}_{s}}$$

### 2.3. Pumping Power

To ensure that nanofluids are evenly distributed throughout the system, an electric pump is required, which consumes energy. It is important to understand the total amount of energy required by the pump to maintain a consistent flow across the collector. In this study, to calculate the power required for the pump, the pumping power calculation formula developed by Garg and Agarwal [39] was used. The pressure drop (Δ*p*), across the collector, can be determined using equation [40].
(15)$$\Delta p=f\frac{\rho V^{2}}{2}\frac{\Delta l}{d}+K\frac{\rho V^{2}}{2}$$

The loss coefficient *K* accounts for the energy losses due to various factors such as entrance effects, exit effects, bends, elbows, valves, etc. This coefficient is often obtained from properties tables that are based on experimental tests or calculated using formulas that do not depend on the density and kinematic viscosity of the heat transfer fluid. The mean flow velocity (*V*) of nanofluids in the system can be calculated using the following equation:(16)$$V=\frac{\dot{m}}{\rho_{hnf}\pi D_{H}^{2}/4}$$
where *D_H_* represents the hydraulic diameter. In the present analysis, *D_H_* = pipe diameter (*d*) was considered.

The frictional factor, *f*, for laminar and turbulent flow, can be expressed as follows [41]: $$f=\frac{64}{Re}\;\mathrm{for}\;\mathrm{laminar}\;\mathrm{flow}$$
$$f=\frac{0.079}{\left(Re^{1/4}\right)}\;\mathrm{for}\;\mathrm{turbulent}\;\mathrm{flow}$$

The Reynolds number is defined by
(17)Re=ρVDHμ

The pumping power can be calculated using the following equation:(18)$$\mathrm{Pumping}\;\mathrm{power}=\left(\frac{\dot{m}}{\rho_{hnf}}\right)\times \Delta p$$

### 2.4. Convective Heat Transfer

The basic forced convective heat transfer model describes the relationship between fluid flow and convective heat transfer in terms of correlations between dimensionless numbers such as Reynolds number (*Re*), Prandtl number (*Pr*), and Nusselt number (*Nu*). For laminar flow inside a pipe with a diameter of *d*, the following theoretically derived relations can be used [42]:(19)$$h_{hnf}=\frac{q}{T_{W}-T_{f}}$$
(20)$$Nu_{hnf}=\frac{h_{hnf}d}{k_{hnf}}$$

The Nusselt number for laminar flow through a circular pipe can be expressed as a function of Reynolds and Prandtl numbers, and can be determined by employing Equation (20) [43],

Nusselt number,
(21)Nu=0.000972Re1.17Pr13 for Re < 2000
where Prandtl number (Pr) can be written as
(22)$$\mathrm{Pr}=\frac{C_{p,hnf}\mu_{hnf}}{k_{hnf}}$$

### 2.5. Thermophysical Properties of Hybrid Nanofluids

The total volume fraction of hybrid nanofluids with conventional fluids is shown in Equation (23) [44].
(23)$$\emptyset_{T}=\emptyset_{np1}+\emptyset_{np2}$$

The density [45], specific heat [46], thermal conductivity [46], and a viscosity [46] of hybrid nanofluids can be presented as follows: (24)$$\rho_{hnf}=\emptyset_{np1}\rho_{np1}+\emptyset_{np2}\rho_{np2}+\left(1-\emptyset \right)\rho_{bf}$$
(25)$$C_{p,hnf}=\frac{\emptyset_{np1}\rho_{np1}C_{p,np1}+\emptyset_{np2}\rho_{np2}C_{p,np2}+\left(1-\emptyset_{T}\right)\rho_{bf}C_{p,bf}}{\rho_{hnf}}$$
(26)$$k_{hnf}=\frac{\frac{\emptyset_{np1}k_{np1}+\emptyset_{np2}k_{np2}}{\emptyset_{T}}+2k_{bf}+2\left(\emptyset_{np1}k_{np1}+\emptyset_{np2}k_{np2}\right)-2\emptyset_{T}k_{bf}}{\frac{\emptyset_{np1}k_{np1}+\emptyset_{np2}k_{np2}}{\emptyset_{T}}+2k_{bf}-2\left(\emptyset_{np1}k_{np1}+\emptyset_{np2}k_{np2}\right)+2\emptyset_{T}k_{bf}}$$
(27)$$\mu_{hnf}=\mu_{bf}\left(\frac{1}{\left(1-\emptyset_{T}{}^{2.5}\right)}\right)$$

## 3. Input Data and Thermal Modelling Process

### 3.1. Input Data

Table 1 and Table 2 are significant sources of data as they present detailed information about the characteristic parameters of the solar collector, the experimental conditions, and the properties of the base fluids and nanoparticles used in the study. These tables offer valuable information for understanding and interpreting the results of the experiment.

**Table 1 nanomaterials-13-01320-t001:** Environmental and analysis conditions for the flat plate solar collector [47].

Parameters of Collector	Value
Type	Black paint flat plate
Glazing	Single glass
Agent fluids	Water, Methanol, Radiator coolant, Ethylene glycol, Engine oil
Absorption area (m^2^)	1.44
Wind speed (m/s)	20
Collector tilt (°)	20
Fluid inlet and ambient temperature (K)	300
Apparent sun temperature (K)	4350
Optical efficiency	84%
Emissivity of the absorber plate	0.95
Emissivity of the covers	0.90
Glass thickness (mm)	4
Insulation thermal conductivity (W/m∙K)	0.06

### 3.2. Solution Procedure

Equations for thermal modelling which were illustrated earlier were solved by Microsoft excel solver and a simple iterative calculation built in excel. In the beginning, the experimental conditions were set for preparing the hybrid nanofluids (the volume fraction ratio of 50%:50% for MWCNT and CuO nanoparticles) [51,52] and the thermophysical properties of hybrid nanofluids were calculated based on Equations (25)–(27). Additionally, the effect of the nanoparticles’ shape and size is considered negligible. Along with the Reynolds number, the Nusselt number and the Prandtl number were estimated, and the pumping power and pressure drop were calculated. Finally, the entropy generation rate, exergy destruction, and exergy efficiency were calculated. 

## 4. Results and Discussion

Figure 1a represented the exergy efficiency against the nanoparticles volume fraction. The efficiency was determined using Equations (10)–(14) and the data in Table 1 and Table 2. As the concentration of nanoparticles increases, there is a slight increase in efficiency for all hybrid nanofluids analysed in this study. Mouli, et al. [53] reported that the exergy efficiency is an increasing function of the nanoparticles’ volume fraction. The figure clearly indicates that the CuO-MWCNT/water hybrid nanofluid exhibits the highest exergy efficiency at all volume fractions compared to other hybrid nanofluids whereas CuO-MWCNT/Methanol hybrid nanofluid shows the lowest efficiency values. At the same volume fraction, the maximum exergy efficiency was found around 7% for CuO-MWCNT/water hybrid nanofluid followed by CuO-MWCNT/radiator coolant (6.7%) CuO-MWCNT/EG (5%), CuO-MWCNT/Engine oil (4%) and CuO-MWCNT/methanol (2.6%) hybrid nanofluids. 

Figure 1b illustrates the exergy efficiency changes with respect to fluid volume flow rate. Similar to Figure 1a, the exergy efficiency increases consistently with an increase in the volume flow rate. The results show that the CuO-MWCNT/methanol hybrid nanofluid exhibits the lowest efficiencies, while the CuO-MWCNT/water hybrid nanofluid has the highest efficiency. This increased efficiency is due to the higher thermal conductivity of the CuO-MWCNT/water hybrid nanofluid compared to the other nanofluids, as reported in reference [54]. Therefore, using CuO-MWCNT/water nanofluid as an absorbent medium in solar collectors can improve exergy efficiency, both in terms of nanoparticle volume fractions and volume flow rate of the hybrid nanofluid in the collector. Hence, CuO-MWCNT/water nanofluid is an excellent choice as an absorbent medium because of its higher exergy efficiency compared to other hybrid nanofluids in this analysis.

Equation (12) was used to calculate the entropy generation (*S_gen_*) using input values from Table 1 and Table 2. The results are presented as entropy generation rate as a function of nanoparticle volume fraction in Figure 2a and as exergy destruction versus nanoparticles volume fraction in Figure 2b. The graph shows that entropy generation decreases as the volume fraction of nanoparticles increases. This phenomenon occurs because an increase in nanoparticle volume fraction leads to an increase in heat flux on the absorber plate, making irreversibility to the dominant factor. Therefore, increasing the volume fraction of nanoparticles results in an increase in nanofluid thermal conductivity [55], which enhances heat transfer capacity; however, the irreversibility reduces the heat transfer rate, which has an adverse impact on entropy generation. Conversely, as the nanoparticle volume fraction decreases, the viscosity of the nanofluid increases and the stiffness of the nanofluid contributes more to the entropy generation. These findings are consistent with those reported by Mahian et al. [56]. From Figure 2, it can be seen that the CuO-MWCNT/water hybrid nanofluid exhibits lower rates of entropy generation compared to CuO-MWCNT/radiator coolant, CuO-MWCNT/EG, CuO-MWCNT/Engine oil and CuO-MWCNT/methanol hybris nanofluids, as expected.

Figure 3 illustrates the variation of entropy generation rate and exergy destruction with the volume flow rate of different hybrid nanofluids using Bejan’s method [11,12]. This analysis is adapted to FPSC’s as minimising entropy generation is crucial for high temperature systems. The entropy generation rate minimisation is equivalent to maximising power output. The method of capturing solar energy is followed by the production of the upstream entropy and downstream from the collector and within the collector. In this study, Ex_in_ is called constant, and the exergy production maximisation (Ex_out_) is the same as the total entropy generation rate minimisation. It is evident from Figure 3 that the thermal entropy generation rate decreases with an increase in the volume flow rate of hybrid nanofluids. The volume flow rate is dependent on the Reynolds number. An increase in the particle volume fraction reduces the thermal entropy generation. The reason for this is that the nanofluids have higher thermal conductivity and the specific heat capacity. Because of increasing heat transfer performance by reducing the volume flow rate of nanofluids reduces thermal entropy generation. From Figure 3 the entropy generation rate is observed as maximum for the CuO-MWCNT/methanol and minimum for the CuO-MWCNT/water nanofluid, hence the exergy output will perceive the maximum for the CuO-MWCNT/water hybrid nanofluid.

The relationship between pumping power and volume flow rate of various hybrid nanofluids is shown in Figure 4a. An increase in volume flow rate results in an increase in mass flow rate, but since density remains constant, the ratio of mass flow rate to density changes. It can be observed from the figure that the pumping power of the engine oil increases more rapidly with increasing volume flow rate than that of other base fluids. This is likely due to the fact that engine oil has a higher density compared to the other base fluids [57]. For example, at the same volume fraction at 0.02, the density of engine oil-based hybrid nanofluids enhancement is found to be 2.5% more than the water-based hybrid nanofluids. 

Equations (15)–(18) describe that the pressure drop is a function of density, velocity, and friction factor for a specific geometric dimension. Figure 4b shows how pumping power varies with volume fraction for different nanofluids used in the study. It is evident from the figure that pumping power remains constant or has a negligible effect with an increase in volume fraction. An increase in the volume fraction results in an increase in both density and mass flow. Therefore, the ratio of mass flow rate and density with respect to volume fractions are the same. The pumping power consequently depends entirely on the pressure drop. The pumping power for CuO-MWCNT/methanol hybrid nanofluids is low compared to other hybrid nanofluids because of their low density and viscosity [58]. From Figure 4b, it is found that methanol-based hybrid nanofluids need the lowest pumping power followed by radiator coolant, water, EG, and engine oil-based hybrid nanofluids. The pumping power remains the same (or has a negligible effect) with an increase in volume flow rate except for engine oil-based hybrid nanofluids. 

The heat transfer coefficient is calculated based on Equation (20). Figure 5a illustrates the heat transfer coefficient varies with the volume flow rate of different hybrid nanofluids. The CuO-MWCNT/methanol hybrid nanofluid shows a higher Nusselt number than the other nanofluids, resulting in a higher heat transfer coefficient. As the volume flow rate increases, the heat transfer coefficient of the CuO-MWCNT/methanol hybrid nanofluid increases exponentially more than that of the other hybrid nanofluids. The increase in the heat transfer coefficients of the CuO-MWCNT/water, CuO-MWCNT/EG, CuO-MWCNT/radiator coolant, and CuO-MWCNT/engine oil nanofluids is significantly lower than that of the CuO-MWCNT/methanol hybrid nanofluid as the volume flow rate increases.

The fluctuation of the heat transfer coefficient with volume percent for various hybrid nanofluids is shown in Figure 5b. According to Equation (20), the Nusselt number and the thermal conductivity of the nanofluid influence the heat transfer coefficient. The Reynolds number and the density of the nanofluid are connected to the Nusselt number. Equation (28), derived from the Hamilton and Crosser model [59], states that the volume fraction and form of the nanoparticles have a direct impact on the nanofluid’s thermal conductivity. The volume fraction has a linear relationship with thermal conductivity [60]. It makes sense that more particles would result in a larger effective surface area for heat transmission. The thermal conductivity of the nanofluid will also improve due to the nanoparticles naturally enhanced thermal conductivity. From our theoretical analysis, the CuO-MWCNT/methanol hybrid nanofluids exhibit the higher values for the Nusselt number. For a fixed intake velocity, Figure 5b indicates that the mass flow rate increases as the volume fractions increase. The increasing rate in the heat transfer coefficient for the CuO-MWCNT/methanol hybrid nanofluid is higher than that of other nanofluids. Besides these, the other nanofluids exhibit almost the same value with the increase in the volume fraction. Recently, researchers [61,62] are suggested to evaluate the performance evaluation criterion and optimisation to identify the optimal trade-off solutions between heat transfer enhancement and pressure drop reduction.

In Figure 6a,b, the Reynolds and Nusselt numbers were calculated using Equations (17) and (20) for various volume fractions. The results in Figure 6b indicate a slight increase in the Nusselt number for all nanofluids as the volume percentage of nanoparticles increases, as well as an increase in the Reynolds number in Figure 6a. The CuO-MWCNT/methanol hybrid nanofluid shows more enhancement compared to other hybrid nanofluids. The increase in the Nusselt number with increasing nanoparticle concentration indicates that the primary cause of heat transfer improvement for a specific Reynolds number is an increase in molecular thermal diffusion. Therefore, the Nusselt number increases with an increase in the Reynolds number. This trend was also observed by Maïga et al. [63].

Figure 7 and Figure 8 depict the correlation between the exergy efficiency, and Nusselt and Reynolds numbers for different volume flow rates and particles volume fraction, respectively. The correlation coefficient for Figure 7 is about 0.974642 and for Figure 8 is about 0.999562 which is very close to 1. Perhaps not surprisingly, the exergy efficiency increases exponentially with the increment of volume flow rates and particle concentrations. The ranges of volume flow rates and particle concentrations were considered 1 to 3.5 lit/min and 0 to 3% respectively for the analysis. CuO-MWCNT/water hybrid nanofluid was chosen as the working medium for their higher performances as discussed earlier. 

Figure 9 shows the changes in exergy efficiency with both volume fraction and solar radiation for CuO-MWCNT/water hybrid nanofluids. The results indicate that the exergy efficiency increases as the volume fraction of both hybrid nanofluids, and solar radiation increases. Specifically, an increase in solar radiation from 400 W/m^2^ to 1000 W/m^2^ results in an exergy efficiency increase of approximately 6% to 17%. This suggests that increasing the volume fraction of nanofluids and solar radiation can lead to improved exergy efficiency, which can be beneficial in various applications such as solar thermal energy conversion. Overall, the findings in Figure 9 highlight the potential for CuO-MWCNT/water hybrid nanofluids to improve the efficiency of energy conversion processes, particularly when used in conjunction with solar radiation. 

A system that is in equilibrium with its surroundings has zero exergy and is said to be at the “dead state” [64]. Figure 10 represents the effects of dead state temperature on exact solar exergy for different solar radiation. As we know, exact solar exergy influences exergy efficiency. Therefore, the characteristic of different parameters relates to the solar exergy that needs to be investigated especially as a function of dead state temperature. Exact solar exergy was calculated using Petela’s theory [65]. The outcome of the hypothesis is shown in Figure 10. As seen in the Figure, the solar exergy reduces with the increase of dead state temperature because of the lower temperature gradient. To improve the solar collector’s performance, it is expected to keep the dead state temperature as low as possible.

The study compared the hybrid nanofluid used in the solar thermal collector to the previously analysed working fluids for different concentrations and flow parameters and presented the results in Table 3. The study found that the energy efficiency of the hybrid nanofluid used in the present study was higher compared to various investigations that used different hybrid nanofluids.

## 5. Conclusions

This study employed the second law of thermodynamics to examine the impact of various nanofluids on the thermal performance and entropy generation of a flat plate solar collector. Additionally, the study evaluated the convective heat transfer properties, pressure drop, pumping power, and flow rate effects of the nanofluids in the solar collector through theoretical analysis. Based on the results obtained from this study, the following conclusions can be drawn:

(a) CuO-MWCNT/water hybrid nanofluid had the lowest entropy generation rate compared to the other nanofluids tested, including CuO-MWCNT/methanol, CuO-MWCNT/engine oil, CuO-MWCNT/radiator coolant, and CuO-MWCNT/EG hybrid nanofluids. Theoretically, this suggests that the CuO-MWCNT/water hybrid nanofluid has the potential to provide the highest exergy output for the FPSC. This finding highlights the potential benefits of using the CuO-MWCNT/water hybrid nanofluid in the design and operation of solar collectors to improve energy efficiency and reduce entropy generation.

(b) CuO-MWCNT/methanol hybrid nanofluid had the lowest pumping power compared to the other base fluids tested, and it also had the highest heat transfer coefficient compared to CuO-MWCNT/water, CuO-MWCNT/engine oil, CuO-MWCNT/radiator coolant, and CuO-MWCNT/EG nanofluids. These findings suggest that the CuO-MWCNT/methanol hybrid nanofluid is more effective in terms of pumping power for the flat plate solar collector. However, the CuO-MWCNT/water hybrid nanofluid is recommended due to its minimum entropy generation behavior. Overall, the study highlights the trade-off between pumping power and entropy generation when selecting a hybrid nanofluid for an FPSC and suggests that the CuO-MWCNT/water hybrid nanofluid may be the optimal choice for maximising energy efficiency while minimising entropy generation.

(c) The addition of suspended nanoparticles in a liquid result in a significant improvement in heat transfer phenomena. Hybrid nanofluids exhibit a higher convective heat transfer coefficient compared to the base liquid at the same Reynolds number. This enhancement in heat transfer is attributed to the higher thermal conductivity and increased surface area of the nanoparticles, which promote efficient heat transfer between the fluid and the heat transfer surface. Overall, the use of nanofluids has the potential to significantly improve the efficiency and effectiveness of various heat transfer applications.

(d) Some assumptions were made in this study to reduce the complexity of the analysis. The nanofluid property, overall heat loss, and solar radiation were considered constant with time. This emphasizes that further study is needed to claim the thermal performance and efficiency improvement of a flat plate solar collector.

## Data Availability

Availability on request, case by case.

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
