# Peer review of "Thermodynamic Analysis of a Flat Plate Solar Collector with Different Hybrid Nanofluids as Working Medium—A Thermal Modelling Approach"

_nanomaterials, 2023, doi:10.3390/nano13081320_

Round 1

Reviewer 1 Report

 This paper investigated the thermodynamic properties of hybrid nanofluid containing CuO/MWCNT with various base fluids, which are useful for a flat plate solar collector. However, the contents in this paper are nothing more than a desk calculation. The reviewer suggests that this paper is unacceptable unless they are compared with required experiments. The important points are as follows:

1. The thermal conductivity and viscosity for nanofluid are difficult to estimate by use of theoretical formula, and they should be measured directly. It is difficult to make sure that the nanofluid is whether Newtonian or non-Newtonian.

2. What was the shape and size of CuO and MWCNT? There is no description in the text.

3. How was the ratio of CuO and MWCNT in the nanofluid?

4. In x-axis, the unit of volume fraction is not [vol%] but [-].

Reviewer 2 Report

This paper presents Thermodynamic analysis of a flat plate solar collector with different hybrid nanofluids as working medium. It is reasonable and complete. But before considering for publication, from reviewer’s view, some revision can be made as follows:

1. The picture and approximate flow chart of the experimental device should be shown in the text, and the accuracy of the experimental data should be verified.

2. Figures a and b in Figure 2 or Figure 3 are basically the same. Is there a direct relationship between the two parameters, and is it necessary to elaborate them separately?

3. On page 11, if the sentence " This is likely due to the fact that engine oil has a higher density compared to the other base fluids " is not rigorous enough. Since it is written in the article, it should be analyzed clearly.

4. It can be seen from Figure 6 that there is no obvious change in Reynolds number and Nusselt number, and the conclusion drawn from this is somewhat farfetched.

5. Some conclusions say that there are similar conclusions in other references, so where is the innovation and whether it is necessary to verify the previous work.

6. Is it necessary to present Tables 3 and 4 in the form of table? It is easy to confuse readers.

7. The numerical change in Figure 10 is not obvious, and the conclusions based on it are not rigorous enough.

8. In conclusion d, some experimental data in this paper has little change, and some assumptions have been made in this paper, which is similar to the ideal working condition. How to ensure the accuracy of the conclusion?

Reviewer 3 Report

In this contribution, predictions to investigate the performances of hybrid nanofluids when applied in flat plate solar collectors, are presented. The predictive model is built up starting from well-known physics law such as second law and so on. The authors investigated five different base-fluids, under different nanoparticles concentrations. From the results, it is shown that CuO-MWCNT/water generally performs better than the other hybrid nanofluids, expect for the heat transfer coefficient. The reviewer thinks that this contribution presents some novel insights about using nanofluids in this particular applications; therefore, it is suggested to consider it for publication after addressing the following points

-          The state-of-art looks quite exhaustive but some improvements are suggested. For instance, there is no mention about nanofluids applied to flat plate solar collector with references to thermal energy storage [doi.org/10.1016/j.applthermaleng.2022.118623;  doi.org/10.1016/j.est.2022.105681]. To help the reader in understanding why is important to investigate this topic, the authors are invited to report all this through the manuscript

-          A sketch of the investigated geometry is strongly suggested; furthermore, did the authors have the chance to compare their results with similar experimental/predictive data from the literature?

-          It is actually not clear how governing equations are solved; for instance, in Eq. 19, how did the authors obtain the heat transfer coefficients from heat flux/temperatures? Are they assuming that one of these is constant/uniform, then the other variable is known from the other balances? Please clarify

-          Which is the difference between Dh (Eq. 17) and d (Eq. 20)? Please write that it is assumed here Dh = d

-          In Eq. 23, the authors report the heat transfer correlation here employed. Why did they decide to use this particular formula, that actually presents a quite high coefficient for the Reynolds number? Furthermore, why did they limit their analysis up to laminar flows (Re < 2000)?

-          In Table 1, it is reasonable to assume that thermophysical properties are not changing with the temperature?

-          Solution procedure is shown in subsection 3.2. Please report some more details; for instance, which iterative method is used? Which is the criterion to assess the algorithm convergence? Furthermore, it is suggested to make references here to the governing equations that have been effectively resolved, i. e. the final equations that set up the equations system

-          In Fig. 5 please correct units for the heat transfer coefficient; by comparing results here with the ones from pumping power, it seems that engine oil is not a good solution. In order to appreciate these features, i. e. which could be a good compromise between heat rate and work in general, the performance evaluation criteria has been widely used [doi.org/10.3390/ma15030968; doi.org/10.1016/j.pnucene.2022.104132]. The authors should make references to all this as a potential alternative solution to evaluate the potential of the solutions that have been proposed here so far

-          In Fig. 6, please use Re and Nu on the y-axis; furthermore, be careful with Nu since it might not be higher than 1 (this happens for engine oil) if the characteristic length is properly chosen

-          Why did the authors use Re*Nu on the x-axis of Fig. 7? Aren't any other ways to show these results by using a different x-axis variable?

-          In Fig. 9, please write the meaning of the variable C at least within the caption to the figure

-          What is the "dead state temperature" in Fig. 10? Please be careful since "date state" is written within the caption to the figure

-          Within the results, some more efforts are required to quantify the results achieved. For instance, why didn't the authors report percentages differences between the different solutions here investigated?

Round 2

Reviewer 1 Report

The reviwer was partialy sutisfied with the response from authors and revised version of paper. However, the reviewer cannot accept the current version which does not include the results under non-Newtonian conditions. Otherwise, the authors should exhibit the evidence that the working fluid behave as Newtonian fluids at current solid fractions in the nanofluid.

Author Response

The response letter is attached. 

Reviewer 2 Report

The revised paper has addressed all my previous comments, I suggest to Accept the paper as it is now.

Author Response

Thank you for accept the manuscript. 

Reviewer 3 Report

The paper can be accepted as it is in the revised form

Author Response

Thank you for accept the manuscript. 

Round 3

Reviewer 1 Report

The reviewer has already been satisfied with the author's response and revised version.